

# Establishment of the non-native horned-face bee *Osmia cornifrons* and the taurus mason bee *Osmia taurus* (Hymenoptera: Megachilidae) in Canada

J. Scott MacIvor[1,2], Charlotte W. de Keyzer[1], Madison S. Marshall[1], Graham S. Thurston[3] and Thomas M. Onuferko[4]

[1] Ecology and Evolutionary Biology, University of Toronto, Toronto, Ontario, Canada
[2] Biological Sciences, University of Toronto Scarborough, Toronto, Ontario, Canada
[3] Canadian Food Inspection Agency, Ottawa, Ontario, Canada
[4] Canadian Museum of Nature, Ottawa, Ontario, Canada

## ABSTRACT

Established populations of the non-native horned-face bee, *Osmia cornifrons* (Radoszkowski, 1887), and the taurus mason bee, *Osmia taurus* Smith, 1873 (Hymenoptera: Megachilidae), have been identified from Canada for the first time. In the US, the importation of *O. cornifrons*, beginning in the 1970s, led to its release for agricultural crop pollination and spread across the country. In this article, we report on *O. cornifrons* captured while sampling wild bees in Toronto, Ontario using hand nets, bug vacuums, and vane traps, as well as established populations in trap nests, from 2017–2020. The morphologically similar *O. taurus*, which was accidentally introduced to the US with shipments of imported *O. cornifrons*, was also recorded in our samples. Recently, a few individual *O. taurus* specimens have been identified from Ontario and Quebec; however, the extent of our sampling included nests, indicating it is also established in Canada. Others have shown its population growth to have been associated with concordant declines in abundances of native mason bee species in the US, and similar impacts are possible in Canada if action is not taken. We propose three non-mutually exclusive possible pathways for the arrival of *O. cornifrons*, as well as *O. taurus*, in Canada: (1) natural migration northward from non-native populations in the US, (2) international importation in the 1980s–2000s to support agricultural research programs, and (3) unintentional release of mason bee cocoons purchased from non-local vendors. We argue that a focus on enhancing populations of locally occurring native bees and stronger policy on the importation and sale of non-native bees are needed.

# INTRODUCTION

Mason bees in the genus *Osmia* Panzer, 1806 (Hymenoptera: Megachilidae) are among the most important solitary bees used in agricultural systems. In particular, the horned-face bee, *Osmia cornifrons* (Radoszkowski, 1887) is valued for pollination of spring-flowering

Corresponding author
J. Scott MacIvor,
scott.macivor@utoronto.ca

fruit trees and shrubs, including apple, blueberry, cherry, and pear (*Bosch & Kemp, 2002*; *Park et al., 2015*). The species is native to Japan, Korea, China, and Far Eastern Russia, where studies have demonstrated it to be an attractive alternative to social western honey bees, *Apis mellifera* Linnaeus, 1758 (Hymenoptera: Apidae), because of its greater pollination efficiency in orchards (*Matsumoto, Abe & Maejima, 2009*). For this reason, *O. cornifrons* was imported into Utah in 1965 where it established populations (*Rust, 1974*), and again in the late 1970s into the northeastern US (primarily Maryland and New York State) for experimentation and release in support of orchard growers (*Batra, 1979*).

The ability of *O. cornifrons* to succeed in new environments is facilitated by the acceptance of a wide range of novel floral resources, and the bee is commonly collected in urban areas where horticultural plantings occur (*Gibbs et al., 2017*). For example, *Vaudo et al. (2020)* evaluated the pollen types collected by *O. cornifrons* using pollen metabarcoding and determined that a variety of plants were visited whose origins span North America, Europe, and Asia.

Cavity-nesting bee species are over-represented among known non-native solitary bees in different regions in the world (*Russo, 2016*). A likely explanation is that cavity-nesting bees prefer to nest in substrates that are more easily and frequently transported by humans (*e.g.*, wood, potted plants, *etc.*) compared to bees with other nesting preferences. *Osmia cornifrons* is a cavity-nesting bee and selects hollow plant stems or holes in wood in which brood cells are constructed in a line toward the apex and partitioned by mud. These bees are gregarious and readily provision brood in trap nests (marketed as bee hotels; *MacIvor, 2017*), which allow practitioners to closely monitor populations, selectively remove parasites, other bees, or wasps, and use applications (*e.g.*, attractant sprays; *Pinilla-Gallego et al., 2022*) to augment the numbers of *O. cornifrons*. The bee has a univoltine life cycle, much of which is spent in a temperature-dependent diapause within a cocoon (*White, Son & Park, 2009*). Individual brood cells can be harvested, inspected, and then stored or easily moved to different fields locally—or globally—to enhance pollination services.

Repeated introductions to the US, establishment, and subsequent trade by vendors, practitioners, and the public have likely played an important role in the spread of *O. cornifrons*. It is importation into the US has additionally resulted in the accidental introduction of a second mason bee species, the taurus mason bee, *Osmia taurus* Smith, 1873, which is morphologically very similar to *O. cornifrons* (*Yasumatsu & Hirashima, 1950*; see also *Gibbs et al., 2017*). *Osmia taurus* individuals have already been recorded in southern Ontario (*Branstetter et al., 2021*) and one individual in Montreal, Quebec (*Normandin et al., 2017*). This is concerning as *LeCroy et al. (2020)* demonstrated an 800% increase in *O. taurus* over a 15-year period in the Mid-Atlantic US beginning in 2003 that was correlated with a 76–91% decline in six other native mason bee species. Non-native bees may compete with native bees for nesting opportunities (*MacIvor & Packer, 2015*), and their populations can grow quickly (*e.g.*, in optimal conditions, *O. cornifrons* populations can increase in size 3–5× per year; *Batra, 1998*). Various non-native bees are also known to frequently pollinate certain invasive plant species, which spread and further impact plant communities that support native bees. For example, non-native western honey bees and the non-native *Megachile apicalis* Spinola, 1808 (Hymenoptera:

Megachilidae) promote the spread of the invasive yellow star-thistle, *Centaurea solstitialis* L. (Asterales: Asteraceae), in California (*Barthell et al., 2001*). *Thornton (2004)* found that non-native *Megachile sculpturalis* Smith, 1853 was a frequent visitor to the invasive kudzu, *Pueraria* DC. (Fabales: Fabaceae), in North Carolina. Non-native bees may also bring with them pests and diseases from their home range that can spread to native species. For example, *Hedtke et al. (2015)* demonstrated that *O. cornifrons* populations in New York were host to chalkbrood (*Ascosphaera* L. S. Olive & Spiltoir (Ascosphaerales: Ascosphaeraceae)), including an associated species from Japan, *Ascosphaera naganensis* Skou, 1988. This pathogen may have spread into populations of the native mason bee *Osmia lignaria* Say, 1837 and be implicated in its recent decline (*LeCroy et al., 2020*). On the other hand, some non-native cavity-nesting bees may have minimal impact on native bee communities. *Megachile rotundata* (Fabricius, 1787), for instance, has been present in North America since the 1940s and is now widespread, but there is no definitive evidence of it competitively excluding native species of *Megachile* Latreille, 1802 (*Pitts-Singer & Cane, 2011*).

In this article, we document for the first time the identification and establishment of *O. cornifrons* in Canada following observational reporting from surveys associated with separate studies using three sampling methods from 2017–2020. We also report an increase in the number of individual specimens of the morphologically similar *O. taurus* collected in Canada. For both species, we present evidence of established populations after uncovering nests from which we reared adults and report on fecundity, parasitism, and mortality. We suggest three non-mutually exclusive possible pathways for the arrival of *O. cornifrons* and expansion of *O. taurus* in Canada. These include (1) natural migration northward from non-native populations in the US, (2) international importation in the 1980s–2000s to support agricultural research programs, and (3) unintentional release of mason bee cocoons purchased from non-local vendors.

## METHODS

Sampling for wild bees in spring occurred across Toronto, Ontario, Canada to answer various ecological questions that are not described here. Sampling was conducted using three primary methods. The first involved the use of trap nests over 6 years spread over two time periods: 2011–2013 and 2018–2020. Trap nests were made of cardboard tubes of three different inner diameter sizes—3.4, 5.5, and 7.6 mm—encased in PVC pipes (see *MacIvor & Packer, 2015* for additional details on design). The second was by targeted surveys in 2017 and 2019 on flowers of eastern redbud, *Cercis canadensis* L. (Fabales: Fabaceae), using hand nets and bee vacuums (2820GA Heavy Duty Hand-held Vac-Aspirator; BioQuip, Rancho Dominguez, CA, USA). The third included the use of blue vane traps (SpringStar Inc., Woodinville, WA, USA) suspended on the branches of eastern redbud trees, before, during, and after redbud bloom in 2018. All sampling occurred in Toronto, the largest and most populated city in Canada, in private yards and seminatural public parks that contained a mixture of meadow and woodland vegetation, all within or surrounded by a heavily urbanized landscape.

*Osmia cornifrons* is markedly distinct from the only two species of *Osmia* subgenus *Osmia* native to North America—*O. lignaria* Say, 1837 and *O. ribifloris* Cockerell, 1900. The females of the two native species have a black (as opposed to yellow) metasomal scopa and lack the distinct clypeal horn-like protrusions present in *O. cornifrons* (*Rust, 1974*). The males of *O. lignaria* and *O. ribifloris* have dark brown/black (as opposed to reddish orange) meso- and metatibial spurs; and their metasomal terga have mixed black and white (as opposed to entirely yellowish) pubescence (*Rust, 1974*). However, given the similarity of *O. cornifrons* to *O. taurus*, with which it has been confused, all sampled specimens were run through the key (and identified following the taxon concepts) of *Yasumatsu & Hirashima (1950)*, with identifications of a subset of males confirmed by dissection and examination of the genitalia, which are markedly different between the two species. Digital focus-stacked images of exemplars were taken at the Canadian Museum of Nature (CMN) using the Leica Z16 APO A apochromatic zoom system combined with the Leica DFC495 and DMC5400 cameras and Leica Applications Suite. These were arranged into labeled figure plates in Adobe Photoshop 2020.

To confirm our identifications based on morphology, a subset of specimens was DNA barcoded. The DNA barcode region, a fragment of the cytochrome *c* oxidase subunit I (COI) mitochondrial gene (*Hebert et al., 2003*; *Hebert, Ratnasingham & de Waard, 2003*), was amplified from template DNA and sequenced at the CMN Laboratory of Molecular Biodiversity in Gatineau, Quebec. From each specimen, the right mid- and hind legs (and in one specimen also the left hind leg) were removed and ground in Bio Plas G-Tube® 2.0 mL Microcentrifuge Tubes containing ~0.2 g of 1.0 mm silicon carbide sharp particles in a BeadBeater. DNA was then extracted from the ground tissues following a protocol adapted from *Ivanova, de Waard & Hebert (2006)*.

The barcode region was amplified using the primers MLepF1 (*Hajibabaei et al., 2006*) and LepR1 (*Hebert et al., 2004*), which were also used for Sanger sequencing in both directions. The PCR reaction mixture (prepared in volumes of 15 or 50 μL) consisted of 7.05 or 29.5 μL ddH$_2$O, 3 or 10 μL 5 × Q5® Reaction Buffer (New England Biolabs (NEB), Ipswich, MA, USA), 0.3 or 1 μL dNTPs (10 mM), 0.75 or 2.5 μL MLepF1 forward primer (10 μM), 0.75 or 2.5 μL LepR1 reverse primer (10 μM), 0.15 or 0.5 μL Q5® High-Fidelity DNA polymerase (0.3 U/rxn) (NEB), and 3 or 4 μL template DNA. For PCR, the following conditions were programed into an Eppendorf® Mastercycler® Pro Thermal Cycler: 1 min at 94 °C, five cycles of 40 s at 94 °C, 40 s at 45 °C, and 1 min at 72 °C followed by 40 cycles of 40 s at 94 °C, 40 s at 51 °C, and 1 min at 72 °C, and 5 min at 72 °C.

The reaction mixture for sequencing consisted of 6.2 μL nuclease-free water, 1.8 μL 5 × ABI buffer, 0.5 μL primer, 0.5 μL BigDye, and 1 μL diluted PCR products for a total of 10 μL per sample. Thermocycling used the following reaction program: 30 cycles of 30 s at 96 °C, 20 s at 50 °C, and 4 min at 60 °C. Sequencing products were transferred to a 96-well plastic plate and therein purified by adding 1 μL 125 μM EDTA, 1 μL 3M NaOAc, and 25 μL 99% ethanol to each sample (to precipitate the DNA) before being subjected to cold centrifugation, rinsing of the DNA pellet in 35 μL 70% ethanol, and additional cold centrifugation. The pellets were thoroughly dried before being resuspended in 15 μL Hi-Di Formamide. The sequencing reaction products were then denatured in a thermal cycler for

5 min at 95 °C, cooled for 2 min, and loaded into an Applied Biosystems® 3500xL Genetic Analyzer for genetic analysis.

ABI trace files were imported into Geneious ver. 11.1.5 (*Kearse et al., 2012*) for assembly into consensus sequences, each of which was assembled from two traces. In the same program, sequences were trimmed, checked for quality, and screened to make sure no stop codons were present in the correct reading frame before being uploaded to the Barcode of Life Data System (BOLD). All newly generated sequences were run through the BOLD Identification Engine (https://www.boldsystems.org/index.php/IDS_OpenIdEngine) for comparison to available barcode sequences for *O. cornifrons* in the BOLD reference library. Genetic distance was visually assessed in a neighbor-joining (NJ) tree generated on the BOLD website based on the Kimura 2 Parameter distance model (*Kimura, 1980*). All ten newly generated barcodes met the criteria to be assigned (in an automated process) a barcode index number (BIN), a unique operational taxonomic unit identifier that usually corresponds with real species limits (*Ratnasingham & Hebert, 2007*, *2013*). Records for the barcoded specimens are maintained in the 'BGTA Bees of the Greater Toronto Area' project on BOLD and will be made public, with the sequences and metadata also released on GenBank (https://www.ncbi.nlm.nih.gov/genbank/), upon publication of this study.

In addition to barcoding select specimens of *O. cornifrons*, previously barcoded exemplars—a female (BOLD sample ID: BIOUG35836-H03) and male (BOLD sample ID: BIOUG05684-A02)—of *O. taurus* deposited in the collection of the Centre for Biodiversity Genomics (CBG) at the University of Guelph in Guelph, Ontario, were personally examined by TO and compared morphologically with specimens collected across Toronto. The female specimen was collected in Washington, D.C., USA in 2017 whereas the male was collected in Port Rowan, Ontario, in 2013.

To determine whether *O. cornifrons* has been reported from Canada previously, we checked existing collections in Manitoba (J.B. Wallis/R.E. Roughley Museum of Entomology in Winnipeg (J. Gibbs, 2021, personal communication)), Ontario (the Canadian National Collection of Insects, Arachnids and Nematodes in Ottawa (by TO in Mar. 2019); the Brock (University) Bee Lab (BBL) in St. Catharines (N. Romero and M. H. Richards, 2021, personal communication); the Packer Collection at York University in Toronto (L. Graham and L. Packer, 2021, personal communication); the Royal Ontario Museum in Toronto (by CdK in Mar. 2019); the University of Guelph Insect Collection in Guelph (S. M. Paiero, 2021, personal communication) and the Raine Collection at the University of Guelph (N. Raine, 2021, personal communication)), Quebec (the Lyman Entomological Museum in Sainte-Anne-de-Bellevue (by TO in Oct. 2021)), and Saskatchewan (the Royal Saskatchewan Museum in Regina (C. S. Sheffield, 2021, personal communication)). We also assessed iNaturalist (www.inaturalist.org) contributions posted to the Global Biodiversity Information Facility (GBIF) (www.gbif.org). No specimens were present at the Canadian Food Inspection Agency (CFIA); however, records of the importation of *O. cornifrons* for pollination research were found and examined.

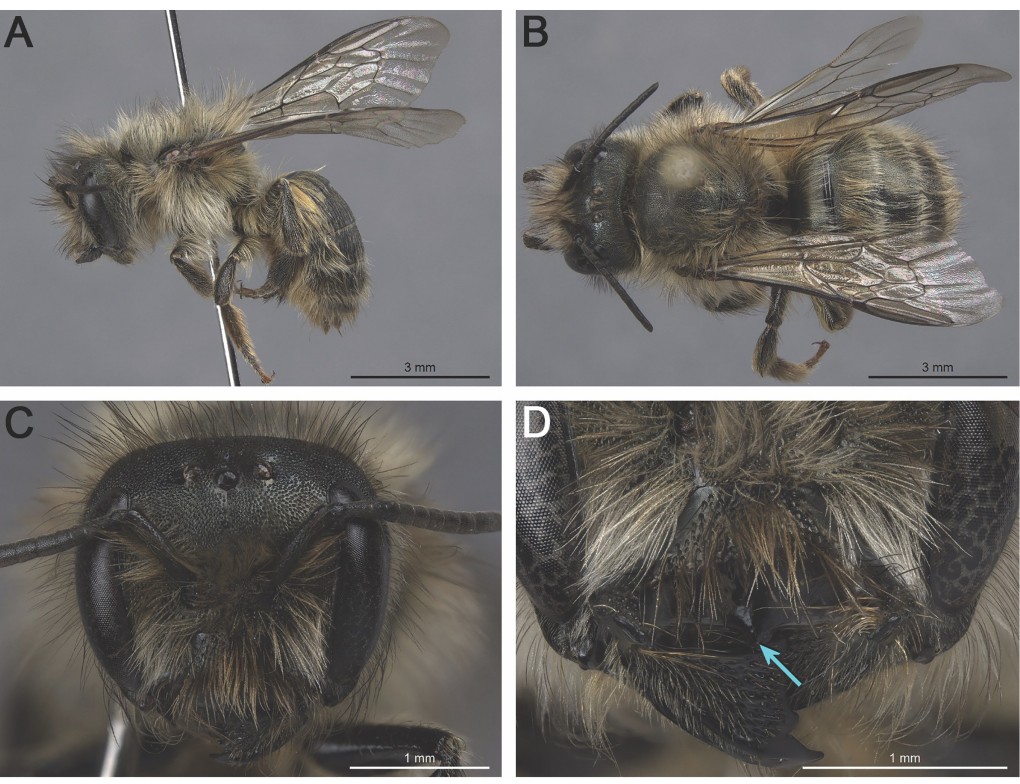

**Figure 1** *Osmia cornifrons* **females: (A) habitus, lateral view (REDBUD-2019-0178); (B) habitus, dorsal view (REDBUD-2019-0227); (C) head, frontal view (REDBUD-2019-0172); (D) head (lower half), frontal view (REDBUD-2019-0167).** The blue arrow in (D) indicates the distinct apicomedian projection of the clypeus.                                         

## RESULTS

Morphological comparisons (see Figs. 1–4 for illustrated differences between *O. cornifrons* and *O. taurus*) and DNA barcodes confirmed the presence of *O. cornifrons* in Canada, with sequenced specimens assigned the BIN BOLD:AAA4494, which is not shared with any other species, including *O. taurus* (BIN BOLD:AAI2020). In total, we collected 65 *O. cornifrons* adults (47 females, 18 males) and 74 *O. taurus* adults (47 females, 24 males, three unsexed specimens) (Materials S1). All individual *O. cornifrons* and *O. taurus* sampled in this study are deposited in the Biodiversity of Urban Green Spaces ('BUGS') Lab at the University of Toronto Scarborough.

The first individual of *O. cornifrons* detected by our sampling was caught on May 24th 2017 in Morningside Park in Toronto by CdK using a hand net (Fig. 5). The female bee was collected from a flowering eastern redbud tree, *Cercis canadensis* (Table 1). Five additional female *O. cornifrons* were subsequently collected between May 7th and May 27th 2018 from blue vane traps suspended from the branches of *Cercis canadensis* at two sites in Toronto. More efforts were directed with bee vacuums on *C. canadensis* trees in 2019, yielding 29 individual female *O. cornifrons* from five sites between May 23rd and May 26th (Fig. 5).

From trap nests, we did not record any *O. cornifrons* in the early years (2011–2013) of our investigations (see *MacIvor & Packer, 2015*). However, from trap nests set out in 2018

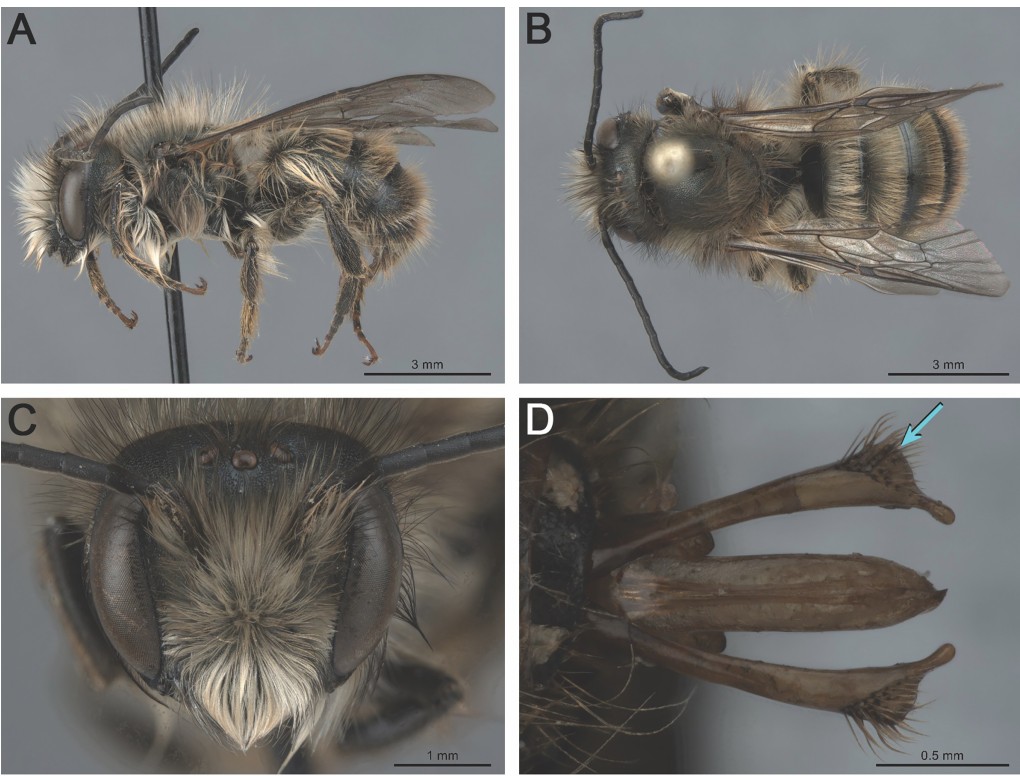

**Figure 2 *Osmia cornifrons* males: (A) habitus, lateral view (MM2-D3-019.020); (B) habitus, dorsal view (MM2-B4-019.020); (C) head, frontal view (MM2-B4-019.020); (D) genitalia, dorsal view (MM146-A2-019-020).** The blue arrow in (D) indicates the distinct subapical lobe-like expansion of the gonocoxite.

we recorded one nest at a single site from which two male adults emerged. In 2019, we recorded 38 brood from six nests at three sites (Table 2). These nests had 27 *O. cornifrons* adults emerge, with 11 larva perishing, three due to parasitism by *Monodontomerus* Westwood, 1833 (Hymenoptera: Torymidae) wasps whereas eight were moldy with the exact cause(s) of death unknown. In 2020, we recorded just one nest containing one individual female that survived to adulthood. In 2019, *O. taurus* was also discovered, in 12 nests recorded from six sites, consisting of 86 brood cells, 67 of which survived to adulthood, with five parasitized by *Monodontomerus* wasps and 14 moldy, having died of unknown causes (Table 2). In 2020, we recorded seven *O. taurus* brood from one nest at a single site, all of which survived to adulthood. *Osmia cornifrons* and *O. taurus* were never recorded from the same site, but *O. cornifrons* was recorded from just two sites, and *O. taurus* from one site, over two consecutive years (Materials S1).

Neither *O. cornifrons* nor *O. taurus* were found in any of the existing collections we surveyed, except for seven individual *O. cornifrons* adults (two females, five males) recorded in 2019 from St. Catharines, Ontario, by the BBL. On May 11[th], 2022, in Windsor, Ontario, colleague S. M. Paiero collected a single female *O. cornifrons* (specimen ID: debu00408282) from flowering eastern redbud at Ojibway Park (42°15′56″N 83°4′41″W)

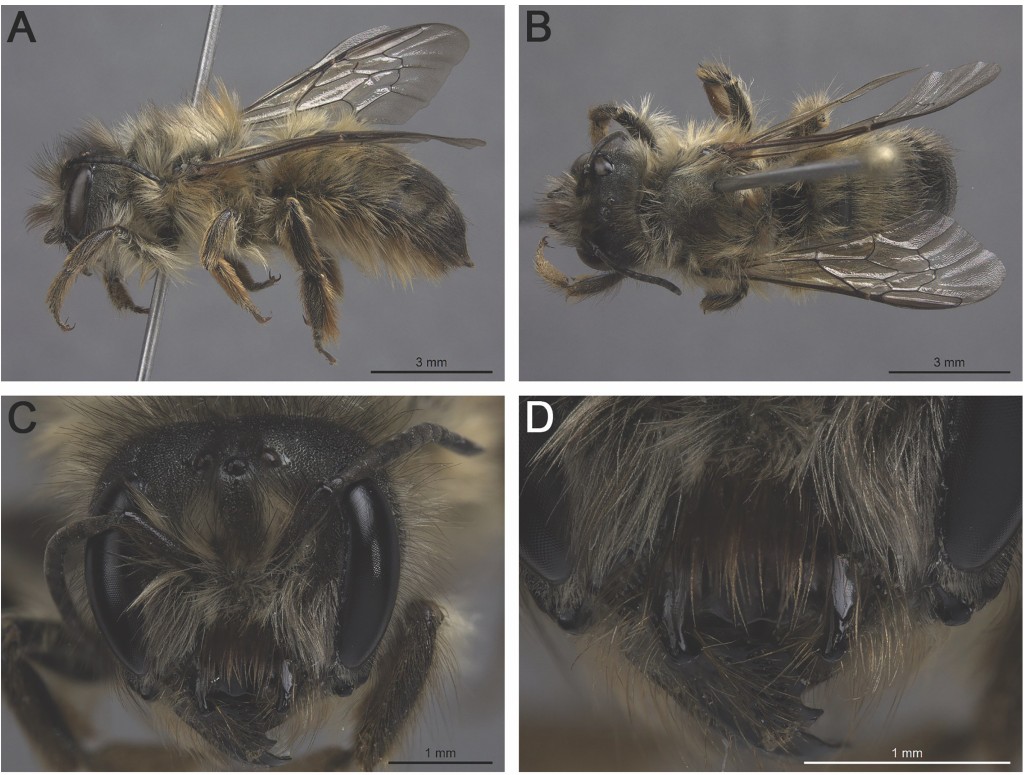

**Figure 3** *Osmia taurus* female (MM110-A5-019.020): (A) habitus, lateral view; (B) habitus, dorsal view; (C) head, frontal view; (D) head (lower half), frontal view.

and two females (debu00408283, debu00408284) on apple (*Malus Mill.* sp. (Rosales: Rosaceae)) at Oakwood Park (42°15′24″N 83°2′2″W). These specimens are deposited at the University of Guelph Insect Collection. We were also able to confidently identify *O. cornifrons* from images associated with the following Canadian records on GBIF: 2236928802 (a female observed on May 7th 2017 in St. Catharines, Ontario), 3118289277 (a female observed on May 17th, 2021 in Burlington, Ontario), and 3302020507 (a female and male in copula observed on April 19th, 2021 in Hamilton, Ontario). One record of *O. taurus* submitted to iNaturalist occurs on GBIF (a female (3333043424) observed on May 10th 2021 from Toronto), whose identity we were able to confirm from images, and DNA barcodes are available (on BOLD) for three specimens collected at Long Point Provincial Park in Ontario in 2013. Additionally, *Normandin et al. (2017)* reported one individual of *O. taurus* from Montreal, Quebec.

Finally, the Canadian Food Inspection Agency determined that *O. cornifrons* had been permitted for research-based study and release, with 15 instances of importation of unknown numbers of specimens for study from 1982–2001 (Table 3). Only in the most recent importation (2001) was it required that each bee be destroyed (and thus not released) following emergence. None of these locations included Toronto, where we report on established populations. Since 2001, there have been an additional 19 permits issued for *Osmia* spp., but none for *O. cornifrons* or *O. taurus* (G. Thurston, 2022, personal

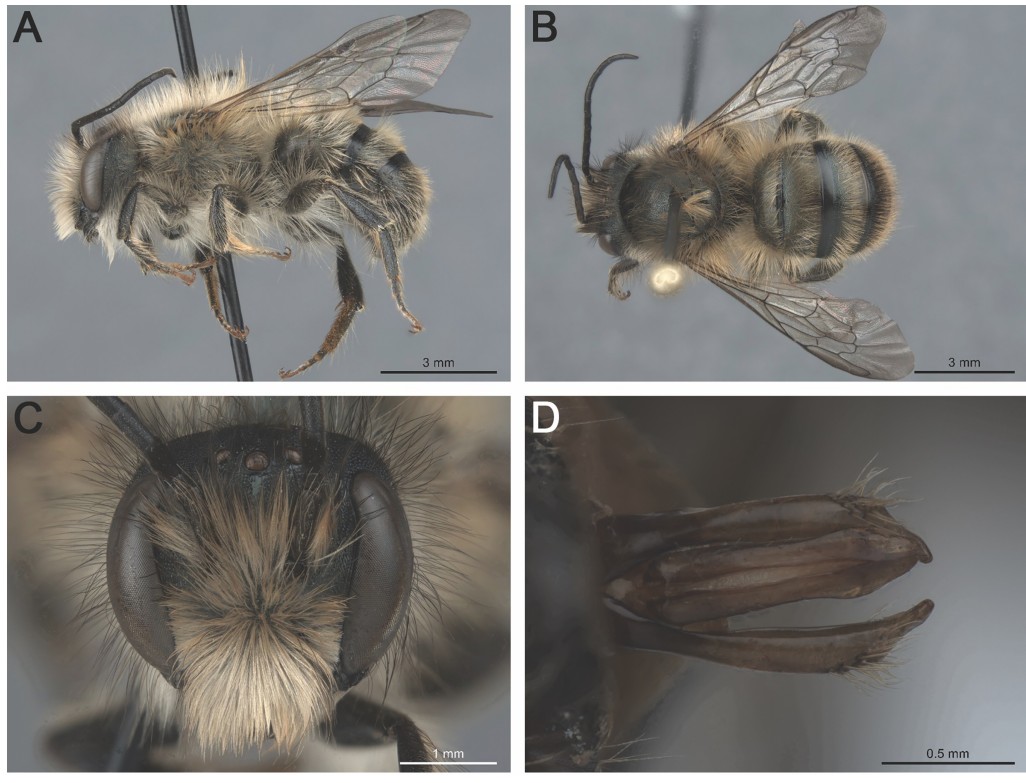

**Figure 4** *Osmia taurus* males: (A) habitus, lateral view (MM36-A3-019.020); (B) habitus, dorsal view (MM36-B6-019.020); (C) head, frontal view (MM36-B6-019.020); (D) genitalia, dorsal view (SU120-C3-019.020).

observations). No restrictions were applied to these permits, and all were for commercial purposes. Fifteen of the permits were for importation to British Columbia, two went to Ontario, and one each to Alberta and New Brunswick.

## DISCUSSION

In this article we document the occurrence of the horned-face mason bee, *Osmia cornifrons*, in Canada for the first time. This cavity-nesting bee is promoted as an effective pollinator in orchards and of other spring flowering crops. Its populations can grow quickly, and the life cycle includes a period of diapause, permitting its storage and transportation to new areas. As a result, the expansion of *O. cornifrons* is a good example of the managed-to-invasion continuum as documented in other bee species imported for agricultural purposes (*Russo et al., 2021*). We also record from our trap nest surveys in 2019 and 2020 the taurus mason bee, *Osmia taurus*, which has been detected in increasing numbers within Canada.

Although no obvious impacts to native bees (*e.g.*, *Osmia* spp.) were determinable from the occurrence records of *O. cornifrons* and *O. taurus* illustrated here alone, these patterns have been documented in the neighbouring US, which should be a cause for concern and call to action for Canadians wishing to preserve native bee diversity. In the following section we discuss possible pathways for how *O. cornifrons*, as well as *O. taurus*, arrived in Canada.
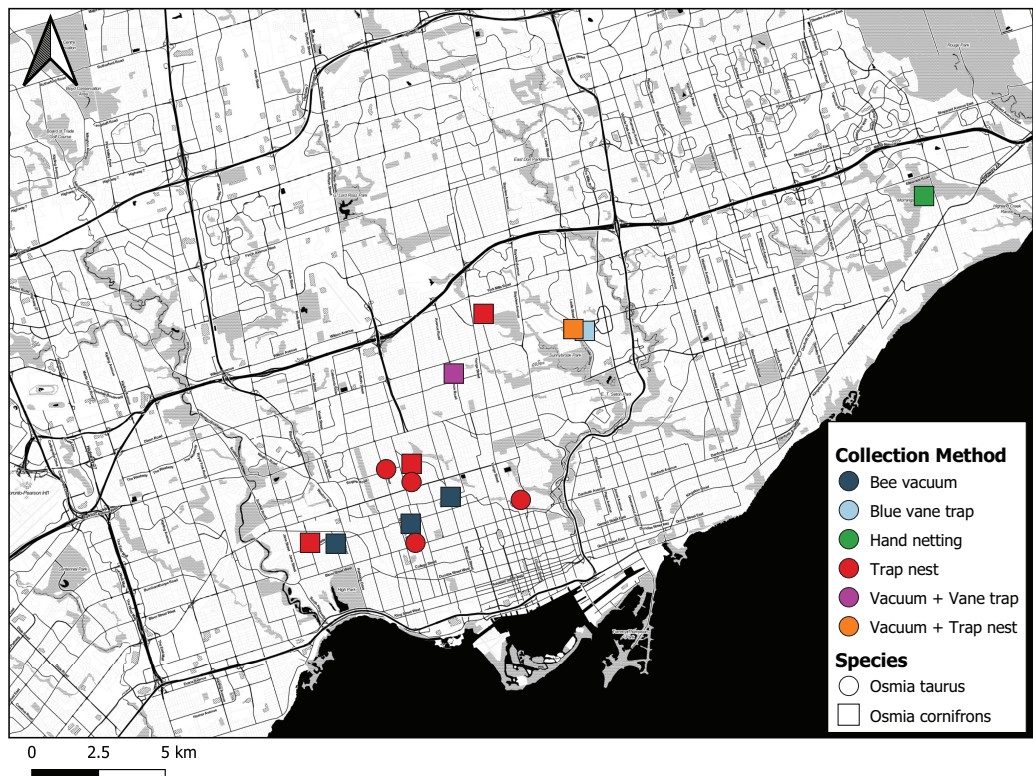

**Figure 5 Map showing where *Osmia cornifrons* and *Osmia taurus* were collected in Toronto and the sampling method used.** Map created by Nicholas Sookhan and published with permission.

**Table 1 Summary data for *O. cornifrons* and *O. taurus* collected in surveys across the City of Toronto from 2017–2020.**

| Sampling method | Year | Species | Sites | # identified | Sex |
|---|---|---|---|---|---|
| Sweep net | 2017 | *O. cornifrons* | 1 | 1 | 1F |
| Vane trap | 2018 | *O. cornifrons* | 2 | 5 | 5F |
| Bee Vacuum | 2019 | *O. cornifrons* | 5 | 29 | 29F |
| Trap nest | 2018 | *O. cornifrons* | 1 | 2 | 2M |
| | 2019 | *O. cornifrons* | 3 | 27 | 11F:16M |
| | | *O. taurus* | 4 | 67 | 44F:20M:3? |
| | 2020 | *O. cornifrons* | 1 | 1 | 1F |
| | | *O. taurus* | 1 | 7 | 3F:4M |
| Total | | *O. cornifrons* | 12 | 65 | 47F:18M |
| | | *O. taurus* | 4 | 74 | 47F:24M:3? |

## Natural migration northward from non-native populations in the US

Because *O. cornifrons* populations are well established in nearby Michigan (*Gibbs et al., 2017*) and New York State (*Russo & Danforth, 2017*), we expected sampling throughout Southern Ontario and published records on GBIF to show this species as more common south of Toronto and nearer to the Canada–US border. Moreover, despite extensive

**Table 2 Number of *Osmia cornifrons* and *O. taurus* nests recovered from Toronto from 2018–2020.** Number of brood cells for each nest, and number of individuals alive and dead (with reason provided) are given.

| Year | Species | Site | Nest # | # Brood | # Alive | Sex | Dead # Parasite | # Moldy |
|---|---|---|---|---|---|---|---|---|
| 2018 | O. cornifrons | a | 1 | 2 | 2 | 2M | 0 | 0 |
| Subtotal | | 1 total | 1 | 2 | 2 | 2M | 0 | 0 |
| 2019 | O. cornifrons | b | 1 | 6 | 3 | 3F | 1 | 2 |
| | | | 2 | 4 | 2 | 2F | 2 | 0 |
| | | c | 1 | 14 | 10 | 3F:7M | 0 | 4 |
| | | | 2 | 1 | 0 | – | 0 | 1 |
| | | | 3 | 8 | 7 | 2F:5M | 0 | 1 |
| | | d | 1 | 5 | 5 | 1F:4M | 0 | 0 |
| Subtotal | | 3 total | 6 | 38 | 27 | 11F:16M | 3 | 8 |
| | O. taurus | e | 1 | 12 | 11 | 7F:4M | 1 | 0 |
| | | | 2 | 12 | 8 | 2F:6M | 3 | 1 |
| | | | 3 | 1 | 0 | – | 0 | 1 |
| | | | 4 | 1 | 0 | – | 0 | 1 |
| | | | 5 | 8 | 4 | 2F:2M | 1 | 3 |
| | | f | 1 | 5 | 5 | 4F:1M | 0 | 0 |
| | | | 2 | 10 | 10 | 7F:3? | 0 | 0 |
| | | | 3 | 9 | 8 | 7F:1M | 0 | 1 |
| | | g | 1 | 1 | 1 | 1F | 0 | 0 |
| | | | 2 | 15 | 8 | 6F:2M | 0 | 7 |
| | | h | 1 | 2 | 2 | 2F | 0 | 0 |
| | | | 2 | 10 | 10 | 6F:4M | 0 | 0 |
| Subtotal | O. taurus | 4 total | 12 | 86 | 67 | 44F:20M:3? | 5 | 14 |
| 2020 | O. cornifrons | i | 1 | 1 | 1 | 1F | 0 | 0 |
| | O. taurus | g | 1 | 7 | 7 | 3F:4M | 0 | 0 |
| Both years | O. cornifrons | 5 total | 8 | 41 | 30 | 12F:18M | 3 | 8 |
| | O. taurus | 4 total | 13 | 93 | 74 | 47F:24M:3? | 5 | 14 |
| Total | | 8 total | 20 | 134 | 104 | 59F:42M:3? | 8 | 22 |

surveying by multiple research groups annually in Southern Ontario, the only *O. cornifrons* collected outside of our own surveys are the seven specimens from St. Catharines in 2019 housed in the collection of the BBL (M. H. Richards, 2022, personal communication) and three from Windsor in 2022 housed at the University of Guelph Insect Collection (S. M. Paiero, 2022, personal communication). It is very possible that the species has migrated into Southern Ontario from populations in the eastern and Midwestern US, but more rigorous surveying in the southern regions of the province is needed. Encouraging people to post photos to iNaturalist for identification is one method for gathering evidence for potential pathways and to detect non-native species entering new regions or countries that might threaten native biodiversity (*Mo & Mo, 2022*). The Canadian records of *O. cornifrons* on GBIF that we were able to confirm from images show this species to be present along the Golden Horseshoe of Ontario, suggesting entry into Canada from the US

**Table 3 Evidence from CFIA requests that *O. cornifrons* was not imported to Toronto but to other areas within Canada.**

| Date | From | To | Conditions |
|---|---|---|---|
| 10/1982 | Beltsville, MD, USA | Campbellville, ON | None listed |
| 03/1983 | Beltsville, MD, USA | Kentville, NS | None listed |
| 06/1983 | Beltsville, MD, USA | London, ON | None listed |
| 01/1984 | Beltsville, MD, USA | London, ON | None listed |
| 02/1984 | Beltsville, MD, USA | London, ON | None listed |
| 02/1987 | Auburn, IN, USA | Guelph, ON | None listed |
| 02/1988 | Auburn, IN, USA | Lasqueti Island, BC | None listed |
| 03/1988 | Beltsville, MD, USA | Ottawa, ON | None listed |
| 03/1989 | Sapporo, Japan | Ottawa, ON | None listed |
| 04/1989 | Auburn, IN, USA | Thornbury, ON | None listed |
| 07/1989 | Auburn, IN, USA | Lasqueti Island, BC | None listed |
| 03/1991 | Auburn, IN, USA | Lambeth, ON | None listed |
| 03/1994 | Portland, OR, USA | Ridgeway, ON | None listed |
| 01/1995 | Beltsville, MD, USA | Chatham, ON | None listed |
| 04/2001 | Auburn, IN, USA | Essex, ON | All imported material must be destroyed following emergence or removal of insects |

*via* the Niagara Peninsula, which is technically an isthmus between Lake Erie and Lake Ontario. This is also suggested by the apparent lack of observations from areas further southwest in the province near the Michigan–Ontario border.

This scenario seems particularly plausible in light of the fact that several non-managed non-native bee species—*Hylaeus punctatus* (Brullé, 1832) in the family Colletidae and *Anthidium manicatum* (Linnaeus, 1758), *A. oblongatum* (Illiger, 1806), *Chelostoma rapunculi* (Lepeletier, 1841), *Hoplitis anthocopoides* (Schenck, 1853), and *Megachile sculpturalis* in the family Megachilidae—were detected in Canada some years after their initial introduction into the eastern United States (*Jaycox, 1967*; *Eickwort, 1970*, *1980*; *Snelling, 1983*; *Smith, 1991*; *Mangum & Brooks, 1997*; *Hoebeke & Wheeler, 1999*; *Mangum & Sumner, 2003*; *Romankova, 2003*; *Buck, Paiero & Marshall, 2005*; *Sheffield, Dumesh & Cheryomina, 2011*). Only a few adventive species, including *Hylaeus communis* Nylander, 1852, *H. pictipes* Nylander, 1852 and *Megachile ericetorum* Lepeletier, 1841, were first detected in North America in eastern Canada and the Great Lakes region, having arrived there presumably *via* the St. Lawrence River, and these (apparently direct) introductions are Western Palaearctic in origin (*Sheffield, Griswold & Richards, 2010*; *Gibbs & Dathe, 2017*; *Martins, Normandin & Ascher, 2017*).

## From former agricultural research-based releases in 1980s–2000s

It is illegal to import live bees from the US into Canada without a permit. To import live mason bees, the requirements are listed in Appendix 1, Section 6 of the CFIA policy D-12-02 (*CFIA, 2022*). A case-by-case evaluation would be made, based on species, origin, intended use, *etc.*, to establish import requirements and/or any other safeguards that may

be applicable. There are very few examples of *O. cornifrons* imported legally (and none for *O. taurus*) due to restrictions permitting only research specimens that must be destroyed upon the completion of any experiments, which eliminates opportunities for their release into local landscapes. Moreover, if these populations were able to establish, given the number of bee surveys completed within Ontario in the last 30 years, it is unlikely such a prolific bee species would evade detection. Thus, past permitted releases are likely not the culprit in the establishment of *O. cornifrons* or *O. taurus* within Canada.

## Purchasing mason bee cocoons from online sellers in the US

Immature cavity-nesting bees have a period of dormancy within cocoons, which allows vendors to mail them with minimal losses of the developing bees due to mortality. These are raised, collected, shipped, and sold to enhance pollination services, traditionally in an agricultural context. This market has recently turned to accommodate smaller numbers of shipped live bee cocoons for home gardeners wishing to supplement purchased bee hotels. However, bees purchased online could arrive from another part of the country (or a different country altogether) and include species different from the one(s) advertised. Movement of wild bees into different regions (and especially urban centres) has increased with internet trade. One can readily purchase wild bee cocoons online, and the high number of observations of both *O. cornifrons* and *O. taurus* in Toronto, the limited observations of *O. cornifrons* in St. Catharines and Windsor, and/or the single observation of *O. taurus* in Montreal (*Normandin et al., 2017*) could represent local releases of non-local bees, perhaps (and possibly, unwittingly) illegally obtained from US sources.

In the US, where there is considerable trade in mason bee cocoons, it is illegal to export live bees to Canada without governmental permission. Federal law regulates the shipment of bees to Canada (7 CFR 322, *Animal and Plant Health Inspection Service, 2004*), but APHIS does not regulate interstate movement of bees within the US. The not-for-profit Orchard Bee Association (OBA) (http://www.orchardbee.org/) has best practices and shipping guidelines for US distributors and does not endorse the movement of mason bee cocoons from eastern to western states. This is because *O. cornifrons and O. taurus* are much more likely to be present in cocoons shipped from the eastern US and could impact western native bee populations. Despite these regulations and guidelines, the simplicity of shipping cocoons by mail necessitates more stringent requirements for those participating in the movement of these bees. This could begin with more resources allocated to public engagement to decrease online trade in cavity-nesting bees and increase gardening and other activities that enhance local, native bee populations. For example, *Seekamp et al. (2016)* showed that an engagement campaign by the U.S. Fish and Wildlife Service led to increased awareness among hobbyist aquarium owners in eight US states of the consequences of internet trade and the proper disposal of invasive species that could threaten the Laurentian Great Lakes region.

## Next steps

The mode of entry to Canada for *O. cornifrons* and *O. taurus* was likely through natural migration across land borders of non-native populations in New York State and Michigan,

but the unintentional release by Ontario residents purchasing mason bee cocoons online is also plausible. Further testing of genetic variation among specimens caught in Ontario could help to determine which of these two possible pathways was mainly if not entirely responsible for the establishment of these species within Canada. The promotion of wild bees (*i.e.*, non-honeybees) as a means for people to engage in efforts to alleviate bee declines may result in misdirection of good intentions (*MacIvor & Packer, 2015*). The inadvertent support of wild bees using improperly evaluated methods such as purchasing mason bee cocoons online and from non-permitted and non-local sources must be highlighted as a potential growing concern for native wild bee conservation.

Given the declines in native mason bee species reported in the Mid-Atlantic US (*LeCroy et al., 2020*), it is imperative that Canadian officials be prepared for possible devastating impacts of increasing numbers of *O. taurus* as well as *O. cornifrons* on the Canadian native bee fauna in the future. The Orchard Bee Association in the US has developed protocols for detecting and managing the spread of mason bees in the country. A similar national association in Canada could play an important role in regulation and education around the movement of mason bee cocoons into different regions, provinces, and territories. At the provincial level, the Government of Ontario has been effective in limiting the spread of wood-boring beetles through public awareness campaigns and by restricting the movement of firewood (*e.g.*, Asian long-horned beetle; *CFIA, 2017*). Similar methods could be employed to restrict mail delivery of mason bee cocoons. Even if marketed as native mason bee cocoons, the possibility of inadvertently including cocoons of *O. cornifrons* or *O. taurus* is not worth the risk of engaging in the practice of purchasing live bees from non-local sources.

Growing numbers of people are purchasing bee hotels to support native bee habitat, and some feel the need to supplement that purchase with live bee cocoons coming from unknown origins. We encourage agricultural suppliers and the public to avoid purchasing bee cocoons from out of province, and especially out of country, for release in conjunction with the placement of bee hotels. Instead, we recommend focusing on habitat stewardship and even the creation of nesting habitat (including bee hotels) to support local pollinators. This could also encourage trapping and raising native mason bee species and other cavity-nesting species for local sales and business development. Campaigns to educate the public on how to identify *O. cornifrons* and *O. taurus* and alert experts to verify identifications of suspected individuals are needed. This is increasingly possible through public data repositories such as iNaturalist, which are frequently checked and used in scientific research (*Mesaglio & Callaghan, 2021*; *Howard et al., 2022*). This will help to manage the spread of potentially invasive bee species and the associated consequences.

## ACKNOWLEDGEMENTS

We thank Terry Griswold for the initial identification of *O. cornifrons* from CdK's collection. Anna Ginter (CMN) is thanked for assistance with molecular work, Dr. Dirk Steinke and Jayme Sones (CBG) for loans of barcoded *O. taurus*, and Nicholas Sookhan for help mapping the sample locations. Additionally, we thank all the curators/collection managers named in the Methods for their assistance with searches through their respective

collections for representatives of *O. cornifrons* and *O. taurus* from Canada, especially Steve M. Paiero, Miriam Richards, and Nora Romero. Finally, we thank three anonymous reviewers for their constructive feedback on our submitted manuscript.

### Funding

Funding support for Madison S Marshall and Charlotte W de Keyzer was provided by a Natural Science and Engineering Research Council of Canada (NSERC) discovery grant awarded to JSM (RGPIN-2018-05660). There was no additional external funding received for this study. The funders had no role in study design, data collection and analysis, decision to publish, or preparation of the manuscript.

### Grant Disclosures

The following grant information was disclosed by the authors:
Natural Science and Engineering Research Council of Canada (NSERC): RGPIN-2018-05660.

### Competing Interests

The authors declare that they have no competing interests.

### Author Contributions

- J. Scott MacIvor conceived and designed the experiments, performed the experiments, analyzed the data, prepared figures and/or tables, authored or reviewed drafts of the article, and approved the final draft.
- Charlotte W. de Keyzer conceived and designed the experiments, performed the experiments, authored or reviewed drafts of the article, checked museum collections for specimens, and approved the final draft.
- Madison S. Marshall performed the experiments, authored or reviewed drafts of the article, and approved the final draft.
- Graham S. Thurston performed the experiments, authored or reviewed drafts of the article, checked government permits for importation and release of bees, and approved the final draft.
- Thomas M. Onuferko conceived and designed the experiments, performed the experiments, analyzed the data, prepared figures and/or tables, authored or reviewed drafts of the article, checked museum collections for specimens, and approved the final draft.

### DNA Deposition

The following information was supplied regarding the deposition of DNA sequences:
The sequences are available at GenBank: ON964443 to ON964452.

## Data Availability

The data are available in the Supplemental Files.

## Supplemental Information

Supplemental information for this article can be found online at http://dx.doi.org/10.7717/peerj.14216#supplemental-information.

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
