# Peer review of "Establishment of the non-native horned-face bee Osmia cornifrons and the taurus mason bee Osmia taurus (Hymenoptera: Megachilidae) in Canada"

_PeerJ, doi:10.7717/peerj.14216_

## Round 0.1 · original submission · Major Revisions

As the manuscript has been meticulously reviewed by three experts in the discipline and after thoroughly going through their reports and glancing at the manuscript, it becomes clear that there are serious technical flaws in the abstract, methodology, insufficient data, and lack of in-depth discussion and localization of results findings that sufficiently suggest to a major revision of the manuscript for publication. Reviewers also suggest adding the map of collection locations. This map should differentiate between sites where nesting populations were found from sites where single adult specimens were collected. A map of the known range of O. cornifrons and O. taurus should also be included. Authors are encouraged to answer all the questions and incorporate the reviewer’s comments.

So, according to the recommendations made by the valuable reviewers, the manuscript needs comprehensive major revisions.

Reviewer 1 ·

Basic reporting

The paper is very well written, uses relevant citations, includes useful and thorough introductory material, and for the most part, includes the validate their findings by molecular identification of exotic bee species. The figures and tables are good and provide supportive information.
I would like to see a citation for lines 347-349 about US federal guidelines concerning shipping of mason bees rom east to west.

Experimental design

This is a good study that documents the occurrence of mason bee species over time in Toronto, Canada. There are diverse trapping methods, multiple localities of collection, and multiple years for comparisons. I would like to see more information about collection sites, e.g., if the landscapes are urban, disturbed, woodlands, natural wildlands, etc.
The addition of the use of molecular tools for species ID validation, the back-checking by expert taxonomists, and the deposition of new genetic data and bee materials in appropriate places is notable and enriches the study.

Validity of the findings

One conclusion is to alert authorities to the potential negative impact of the exotic Osmia on native Osmia. Yet, there are no data on the relative abundance of O. lignaria or O. ribifloris, etc. in this paper. How many of the native species were co-collected in all the trap types? Did this change over time (reduced number of native species)? It is further important to know if the exotic species occur in the natural areas where native species are found, such that the concern for the native bees might be less if they are only being potentially reduced (for the possible reasons given) in human or disturbed habitats.
Also, another example is the exotic Megachile rotundata that is managed and produced at large scales in Canada for alfalfa seed and hybrid canola pollination. But there have been no documentations of their negative effects on native Megachile spp. (if such studies have ever been done). Just another example that could be discussed here, too, especially because they are sold by Canadian bee producers in the millions to US farmers in the western US.

Reviewer 2 ·

Basic reporting

Overall, the manuscript is well written and easy to follow. There are a few areas that should be addressed which are included below. After reading the article I am left wondering if collecting 60-70 specimens over the time span of the survey constitutes labeling these species as “established.” I recommend the authors should review the following article:
Kocovsky, P., M. Sturtevant, R. Schardt, J. 2018. What it is to be established: policy and management implications for non-native and invasive species Management of Biological Invasions, 9(3), 177-185, DOI http://dx.doi.org/10.3391/mbi.2018.9.3.

Authors do not present findings that demonstrate any negative impacts from limited establishment by these species within the region and should be mindful of language used to describe potential threats posed to native bee populations.
Abstract and line 30 - “Its population growth has been associated with concordant declines in abundances of native mason bee species in the US.” This should either be cited or reworded.

Line 57 - “Cavity-nesting bees are over-represented among known non-native solitary bees in different regions in the world (Russo, 2016). Please provide greater detail. For example: Cavity-nesting bees are over-represented among known non-native solitary bees in different regions in the world as a high number of trap nest samples are analyzed yearly when compared to bee with other life histories.

Line 81 - “Non-native bees also pollinate non-native plant species, which spread and further impact native plant communities”. Many native bee species also pollinate non-native plant species. Reword.

“Non-native bees also bring with them pests and diseases from their home range that can spread to native species. For example, Hedtke et al. (2015) demonstrated O. cornifrons populations in New York were host to chalkbrood (Ascosphaera), including an associated species from Japan, Ascosphaera naganensis Skou.” Should read “Non-native bees may also…..” Native species of Osmia are also host to Ascosphaera.

Experimental design

Research question not well defined and should be clearly stated within the manuscript. This is an observational report of finding non-native species in samples from another set of studies.

Missing from this manuscript is a map of collection locations. This map should differentiate between sites where nesting populations were found from sites where single adult specimens were collected. A map of the known range of O. cornifrons and O. taurus should also be included. Lastly, I wonder if any attempt has been made to include surveys of commercially available mason bee stocks. In the Pacific Northwest we believe that this was may have been an important pathway for distribution of cornifrons.

Validity of the findings

Although findings are relevant and should be reported, the scope of survey work does not allow for the authors to answer questions pertaining to the introduction of these species to the region. I recommend sections covering hypothesis be removed from the manuscript or rephrased and included in the discussion section. Authors state that permits were issued for the use of O. cornifrons without restriction in Canada, implying that introductions likely did occur. It is stated that O. cornifrons and O. taurus were recorded at only one site in consecutive years, implying that these species may not be establishing at high rates. This may imply that absence of detection at research sites where O. cornifrons was permitted does not rule out importation as a pathway.

Reviewer 3 ·

Basic reporting

This study reported the identification and establishment of O. cornifrons in Canada and suggested and discussed three hypothetical pathways for the arrival of O. cornifrons. This is a timely important report for the future distribution and invasion of O. cornifrons and associated parasites including Monodontomerus wasps. I have known that O. cornifrons has been found in Canada (Ontario) in the past, but it is surprising that no report has been made on the current status of O. cornifrons in Canada so that the authors can claim in lines 93-94 "... we document for the first time the identification and establishment of O. cornifrons in Canada."

Experimental design

In addition to collecting, identifying, and reporting O. cornifrons in Canada, the authors suggested and kind of tested three potential pathways of O. cornifrons arrival to Canada. However, this study was not specifically designed for fully testing the three potential pathways. When I read the last paragraph of the Introduction section mentioning "pathways", I thought that the authors conducted population genetic analyses to determine the origin or variations (due to multiple introductions to Canada) of O. cornifrons and O. taurus populations. As there are no hard genetic evidence, and only relies on limited survey data, the authors' conclusions are still a bit speculated although the authors did a fair job answering their potential pathways one by one in the Discussion section. I wonder if the Canadian O. cornifrons populations are from the eastern U.S. only, and how much genetic variation they have in Canada. I suggest adding this or similar context in future studies needs to be done in the Discussion section.

In terms of sampling/surveying methods, the authors stated three methods: trap nests, targeted surveys on C. canadensis flowers, and blue vane traps. As the authors did not provide any details (e.g., when, where, how many, how long, or even how they did a survey of O. cornifrons on flowers), it is impossible for others to replicate the study. Also, it is impossible to know what percentage of samples had O. cornifrons and O. taurus. So, the authors need to add more detailed information about their sampling/survey methods.

Validity of the findings

The authors claimed "we document for the first time the identification and establishment of O. cornifrons in Canada" in lines 93-94. I am not sure if the authors can claim that this study is the first identification of O. cornifrons in Canada. Please make sure if the authors can claim that. Also, the authors need to be careful when they state "establishment". In biological invasion, the establishment needs to be defined. I am not sure how the authors defined the "establishment" of O. cornifrons in Canada, so adding how the authors can claim or define the establishment of O. cornifrons. All of these need to be addressed in the Discussion section.

Additional comments

Are Monodontomerus wasps mentioned in the manuscript also invasive species from the U.S.? It would be interesting to add to the Discussion section although not necessary.

---

## Round 0.2 · accepted · Accept

I have gone through the revised manuscript in detail, All the comments that reviewers had said has been incorporated accordingly. Now the manuscript is ready for publication and the decision is accepted